# FAST-DLLM V2: EFFICIENT BLOCK-DIFFUSION LLM

**Chengyue Wu[1,2]    Hao Zhang[2]    Shuchen Xue[2]    Shizhe Diao[2]    Yonggan Fu[2]    Zhijian Liu[2]**
**Pavlo Molchanov[2]    Ping Luo[1]    Song Han[2,3]    Enze Xie[2]**
[1]The University of Hong Kong    [2]NVIDIA    [3]MIT

https://nvlabs.github.io/Fast-dLLM/v2/

## ABSTRACT

Autoregressive (AR) large language models (LLMs) have achieved remarkable performance across a wide range of natural language tasks, yet their inherent sequential decoding limits inference efficiency. In this work, we propose Fast-dLLM v2, a carefully designed block diffusion language model (dLLM) that efficiently adapts pretrained AR models into dLLMs for parallel text generation—requiring only ∼1B tokens of fine-tuning. This represents a 500× reduction in training data compared to full-attention diffusion LLMs such as Dream (580B tokens), while preserving the original model's performance. Our approach introduces a novel training recipe that combines a block diffusion mechanism with a complementary attention mask, enabling blockwise bidirectional context modeling without sacrificing AR training objectives. To further accelerate decoding, we design a hierarchical caching mechanism: a block-level cache that stores historical context representations across blocks, and a sub-block cache that enables efficient parallel generation within partially decoded blocks. Coupled with our parallel decoding pipeline, Fast-dLLM v2 achieves up to 2.5× speedup over standard AR decoding without compromising generation quality. Extensive experiments across diverse benchmarks demonstrate that Fast-dLLM v2 matches or surpasses AR baselines in accuracy, while delivering state-of-the-art efficiency among dLLMs—marking a significant step toward the practical deployment of fast and accurate LLMs.

## 1 INTRODUCTION

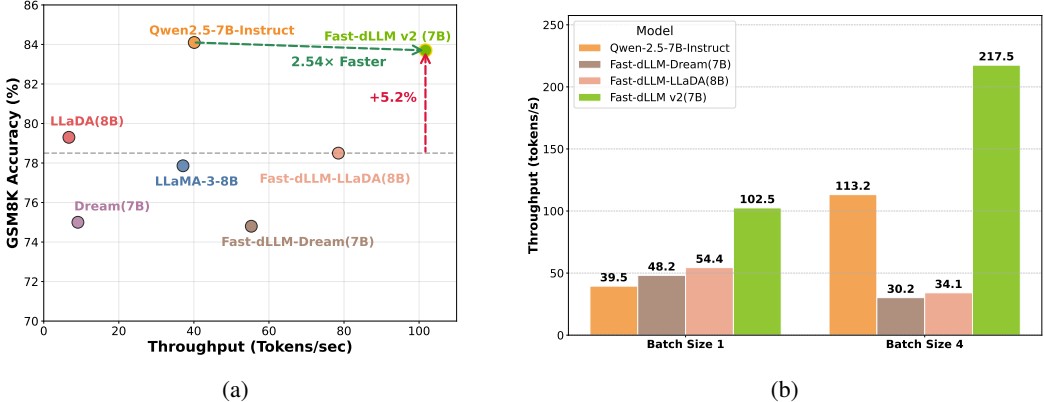

Figure 1: **Performance comparison of Fast-dLLM v2. (a)** Comparison of throughput and GSM8K accuracy among baseline models and the Fast-dLLM variants in A100. Fast-dLLM v2 (7B) achieves 2.54× higher throughput than Qwen2.5-7B-Instruct while offering comparable accuracy. Additionally, it improves accuracy by +5.2% over Fast-dLLM-LLaDA, which is based on optimized LLaDA. **(b)** Throughput comparison under different batch sizes. Fast-dLLM v2 significantly outperforms all baselines at both batch size 1 and 4, demonstrating superior scalability and efficiency.

Recent years have witnessed autoregressive (AR) large language models (LLMs) (Radford & Narasimhan, 2018; Radford et al., 2019; Brown et al., 2020; OpenAI, 2022) achieving remarkable

performance across a wide range of natural language tasks. Their capacity to generate fluent, coherent text by modeling next-token prediction has made them the prevailing paradigm in most deployed systems. However, AR models suffer from inherent inefficiencies: Since tokens are generated one-by-one in a strict left-to-right order, they cannot exploit full parallelism during decoding.

On the other hand, diffusion-based language models (dLLMs) (Google DeepMind, 2025; Inception Labs, 2025; Zhu et al., 2025; Ye et al., 2025a) offer a promising alternative. By allowing multiple tokens (or even entire blocks of tokens) to be predicted or refined jointly, dLLMs can in principle achieve much higher decoding parallelism. Nevertheless, in practice, they come with their own significant drawbacks: they often cannot use KV cache effectively due to bidirectional attention, their inference latency often exceeds that of AR models, and many require fixed sequence lengths or have restricted flexibility in generation length. These limitations have prevented diffusion-based models from outperforming AR models in speed while maintaining comparable quality. Some works employ approximate KV cache mechanisms to reuse computation, such as the DualCache in Fast-dLLM (Wu et al., 2025). However, this does not fundamentally resolve incompatibility of dLLMs with KV cache, since such approximate caches are not equivalent to the original computation.

To bridge these paradigms, block diffusion language model (as in BD3-LMs (Arriola et al., 2025)) has been proposed: it interpolates between purely autoregressive and diffusion regimes by generating tokens in blocks, performing diffusion within each block, while conditioning on previous blocks autoregressively. BD-LMs achieve two desirable properties: flexible sequence length (arbitrary or variable length generation), and KV caching between blocks, enabling improved inference efficiency. However, BD3-LMs have so far only been validated on relatively small-scale models and conventional LM metrics, rather than modern large-scale LLM settings. As such, their practical applicability to state-of-the-art LLMs remains unclear, especially in terms of maintaining high-quality text generation and robust scaling behavior.

In this work, we propose Fast-dLLM v2, a carefully designed block diffusion language model (dLLM) that transforms pretrained autoregressive (AR) models into diffusion-style decoders for parallel text generation. Unlike prior block diffusion approaches that remain limited to small-scale validation, Fast-dLLM v2 is explicitly built to scale to large LLMs and real-world tasks. A key feature of Fast-dLLM v2 is its data efficiency: while full-attention diffusion models such as Dream (Ye et al., 2025a) require on the order of 500B tokens for fine-tuning, our method adapts AR models into block diffusion models with only about 1B tokens of fine-tuning—achieving lossless adaptation without retraining from scratch. Unlike the full-attention dLLM in Dream, our design uses a block-wise attention mask structure closer to the original AR models, making the adaptation process inherently more compatible and data-efficient. Our method further introduces a novel training recipe that combines a block diffusion mechanism with a complementary attention mask, enabling block-wise bidirectional context modeling while simultaneously preserving the original AR training objectives and predictive performance. To enhance inference speed, we design a hierarchical caching mechanism: a block-level cache that stores historical context representations across blocks, and a sub-block cache that supports efficient parallel decoding within partially generated blocks which adopts the DualCache in Fast-dLLM (Wu et al., 2025).

In consequence, Fast-dLLM v2 achieves up to 2.5× speedup over standard AR decoding without compromising generation quality. Extensive experiments across diverse tasks confirm that Fast-dLLM v2 not only matches the accuracy of AR baselines but also achieves state-of-the-art efficiency among diffusion-based LLMs, marking a significant step toward the practical deployment of fast and accurate language models. In summary, our contributions are threefold:

1. We identify the AR-friendly nature of our block-wise attention design and leverage it to present a post-training strategy for adapting pretrained AR models into block-diffusion frameworks, requiring only affordable fine-tuning rather than full retraining. Specifically, Fast-dLLM v2 achieves lossless adaptation with just ~1B tokens, compared to ~500B tokens required by Dream (Ye et al., 2025a).

2. We introduce an inference strategy that combines a hierarchical caching mechanism with block-wise parallel decoding. This design enables effective reuse of context across blocks and accelerates token generation within each block, yielding substantially faster inference than prior diffusion-based methods.

3. We conduct comprehensive large-scale experiments on models up to 7B parameters and diverse tasks, showing that Fast-dLLM v2 achieves up to $2.5\times$ speedup over standard AR decoding while maintaining comparable generation quality.

## 2 RELATED WORK

### 2.1 MASKED DIFFUSION LLM

Initial work by (Sohl-Dickstein et al., 2015; Hoogeboom et al., 2021) first pioneered the use of diffusion models for discrete data. This concept was later generalized by D3PM (Austin et al., 2021) using a forward process defined as a discrete-state Markov chain with general transition matrices $Q_t$. CTMC (Campbell et al., 2022) extended this to continuous time, while SEDD (Lou et al., 2023) instead modeled the likelihood ratio $\frac{p_t(y)}{p_t(x)}$ using Denoising Score Entropy. Masked Diffusion Models (MDMs)—also called absorbing state discrete diffusion—are prominent in discrete diffusion models. During training, MDMs randomly replace tokens with a special [MASK] token according to mask ratio $t$, where $t \in [0, 1]$ interpolates between $x_0$ ($t = 0$) and a fully masked sequence ($t = 1$). The MDMs have been scaled up to 7B level, with LLaDA (Nie et al., 2025) being trained from scratch on the MDM loss, and Dream (Ye et al., 2025b;a) being adapted from the existing Qwen-2.5 7B (Qwen et al., 2025).

### 2.2 INTERPOLATION BETWEEN AUTOREGRESSIVE AND MASKED DIFFUSION

Several recent works have explored block-wise diffusion for non-autoregressive text generation. SSD-LM (Han et al., 2022) introduced a block formulation of Gaussian text diffusion. Building on this, AR-Diffusion (Wu et al., 2023) extended SSD-LM by incorporating a left-to-right noise schedule. For masked diffusion models, BD3-LM (Arriola et al., 2025) interpolates between discrete denoising diffusion and autoregressive models by using inner-block diffusion within a global left-to-right structure. Concurrent to our work, SDAR (Cheng et al., 2025) successfully finetuned a block diffusion model from a pretrained autoregressive model. While sharing the conceptual direction of AR-to-diffusion adaptation, our work diverges by establishing a systematic framework designed for high data efficiency and decoding robustness. Specifically, we introduce *sub-block decoding* to resolve the performance degradation caused by training-inference block size mismatches, and employ *complementary masking* to maximize the denoising signal within blocks. D2F (Wang et al., 2025b), inspired by Diffusion Forcing (Chen et al., 2024) and CausVid (Yin et al., 2025), distilled a large diffusion language model (dLLM) into a more efficient block diffusion model. Set Block Decoding (Gat et al., 2025) integrates standard next-token prediction (NTP) and masked token prediction (MATP) within a single architecture to enable the generation of multiple tokens simultaneously. Compared to these concurrent works, Fast-dLLM v2 is distinguished by its data-efficient fine-tuning process, requiring only 1B tokens.

### 2.3 ACCELERATION FOR DIFFUSION LLM

Recent research has focused on accelerating the inference of diffusion language models, primarily through two avenues: caching mechanisms and advanced decoding strategies. A significant bottleneck in dLLM inference is the computational cost associated with the bidirectional attention mechanism. To address this, several caching techniques have been proposed. FAST-DLLM (Wu et al., 2025) proposed DualCache. This method caches the KV activations for both the preceding text (prefix) and the subsequent masked tokens (suffix). dKV-Cache (Ma et al., 2025) proposed a delayed caching strategy. dLLM-Cache (Liu et al., 2025) accelerates inference by combining prompt caching with an adaptive partial response cache that decides whether to reuse or recompute the generated prefix at each step to balance efficiency and accuracy. Sparse-dLLM (Song et al., 2025) accelerates inference by using the model's attention scores to dynamically drop unimportant tokens from the Key-Value cache. DPad (Chen et al., 2025) proposed to restrict the attention mechanism to a small, fixed-size window of recent suffix tokens.

As for advanced decoding strategies, FAST-DLLM (Wu et al., 2025) proposed an adaptive confidence-based parallel decoding algorithm. EB-Sampler (Ben-Hamu et al., 2025) introduces a simple drop-in replacement for existing samplers, utilizing an Entropy Bounded unmasking proce-

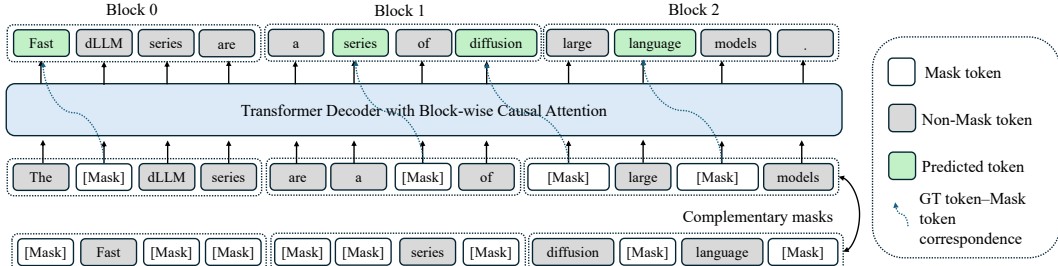

Figure 2: Training process of Fast-dLLM-v2. The input sequence is decoded block by block. Within each block, the model performs next-token prediction with partial masking. To ensure every token is trained, complementary masks are introduced so that masked tokens in one view can be predicted from the other. We only apply loss to predicted tokens that are highlighted in green, and dashed curves connect Mask tokens to their corresponding predictions.

dure that dynamically unmasks multiple tokens in one function evaluation with predefined approximate error tolerance. Dimple (Yu et al., 2025) proposed confident decoding, which dynamically adjusts the number of tokens generated at each step. WINO (Hong et al., 2025) employs a parallel draft-and-verify mechanism, aggressively drafting multiple tokens while simultaneously using the model's bidirectional context to verify and re-mask suspicious ones for refinement. SlowFast Sampling (Wei et al., 2025) proposed a dynamic sampling strategy that adaptively alternates between exploratory and accelerated decoding stages. LaViDa (Li et al., 2025c) tests timestep shift for efficient sampling. Wang et al. (2025a) proposed Temporal Self-Consistency Voting that aggregates predictions across denoising steps to select the most consistent output. Prophet (Li et al., 2025b) dynamically decides whether to continue refinement or decode all remaining tokens in one step.

Beyond accelerating diffusion models themselves, block diffusion is also compatible with Auto-Regressive (AR) acceleration techniques, such as speculative decoding. DiffuSpec (Li et al., 2025a) demonstrates that a small diffusion model (e.g., Dream-7B) can serve as an effective draft model for speculative decoding, achieving a $3.1\times$ speedup with AR models like Qwen2.5-32B. Given that FAST-DLLM V2-7B is nearly $10\times$ faster than Dream-7B (as shown in Figure 1a), employing it as a draft model within similar frameworks offers a promising avenue to further enhance decoding efficiency in future work.

## 3 METHODOLOGY

### 3.1 PRELIMINARY

Let $x = \{x^1, x^2, \ldots, x^L\}$ denote a token sequence of length $L$. Traditional autoregressive models generate $x$ sequentially by modeling the conditional distribution $P_\theta(x^i|x^{<i})$, and are trained to minimize the cross-entropy loss.

In contrast, diffusion language models define a generative distribution via a forward noising process and a learned reverse denoising model. At time $t \in (0, 1)$, each token in $x_0$ is masked independently with probability $t$, producing a corrupted sequence $x_t$. The reverse model $p_\theta(x_0|x_t)$ predicts the original tokens given the noised input.

The model is trained to minimize the expected masked token prediction loss:

$$\mathcal{L}(\theta) = -\mathbb{E}_{t,x_0,x_t}\left[\frac{1}{t}\sum_{i=1}^{L}\mathbf{1}[x_t^i = \texttt{[MASK]}]\log p_\theta(x_0^i \mid x_t)\right],$$

where $t \sim \text{Uniform}(0, 1)$ and $x_t$ is sampled from the forward process. The loss is computed only over the masked tokens.

### 3.2 ADAPTATION TO BLOCK DIFFUSION LLM

We build our block-wise diffusion training pipeline on top of pretrained Qwen2.5-Instruct models (Qwen et al., 2025), including both 1.5B and 7B variants. Fine-tuning is conducted as super-

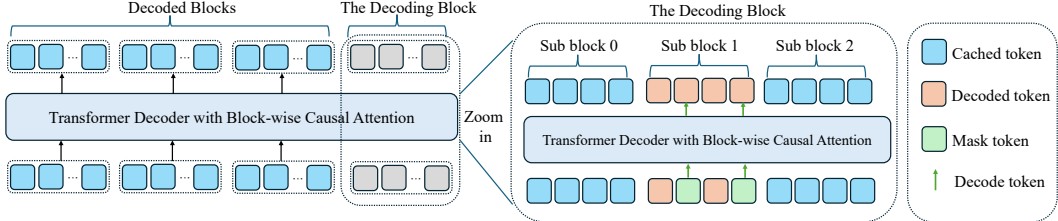

Figure 3: Illustration of the inference process. The sequence is decoded block-by-block. The decoded blocks are cached to speed up inference. Within each block, we adopt the parallel decoding and DualCache in Fast-dLLM to further accelerate inference.

vised fine-tuning (SFT) on instruction-tuning data, where each training batch is constructed using our blockwise diffusion setup. Specifically, we introduce partial token masking within each block together with a complementary masking strategy (Li et al., 2025c) to ensure that every token is trained in both visible and masked contexts. The overall architecture and training workflow are illustrated in Figure 2.

**Block-wise organization.** Given a set of tokenized samples, we first pad each sequence to a length that is an integer multiple of the block size $D$ by appending [MASK] tokens as needed. These padding tokens are ignored in the loss computation and do not contribute to gradient updates. After this alignment step, we pack the padded sequences by concatenating them into a long token stream and splitting it into training sequences of a fixed context length $L$. Each packed sequence is therefore naturally divided into $B = L/D$ non-overlapping blocks of size $D$, already aligned by construction. This block-aligned packing ensures efficient batching while avoiding block boundaries crossing sample boundaries.

**Masked token prediction with complementary views.** For each block, we randomly sample a binary mask $m \in \{0,1\}^D$, where $m_j = 1$ means position $j$ is replaced with a learned [MASK] embedding. To ensure all tokens receive both masked and unmasked supervision across training, we use a *complementary masking* strategy: each training sample is duplicated into two *views* with masks $m$ and $\bar{m} = 1 - m$. These two views are placed together in the same batch, so the model can jointly see masked and unmasked contexts across the views.

**Token shift for prediction.** To preserve the pretrained AR model's representation quality (Ye et al., 2025b;a), we adopt a shifted-label strategy: prediction of a token at masked position $i$ uses the logit from its preceding position $(i - 1)$. Concretely, if $x_i$ is masked, the model uses the hidden state at $i - 1$ to predict $x_i$, consistent with the next-token prediction mechanism in causal language models. This allows dLLM to maintain AR-like temporal representations while supporting intra-block diffusion.

**Training objective.** We minimize the masked-token-only cross-entropy loss:

$$\mathcal{L}_{\text{block}}(\theta) = -\mathbb{E}_{x,m}\left[\sum_{i=1}^{L} \mathbf{1}[x_t^i = \text{[MASK]}] \log p_\theta(x_0^i \mid x_{<i}, x_{\text{block}(i)})\right],$$

where $x_{\text{block}(i)}$ denotes all tokens in the block containing position $i$ (including masked/unmasked), and $x_{<i}$ are clean tokens from earlier blocks.

**Block-wise attention masking.** We use a hybrid attention scheme similar to that in (Arriola et al., 2025). For each training sample, we concatenate the noised sequence $x_t$ and its corresponding clean sequence $x_0$ along the sequence dimension, resulting in a total length of $2L$. The attention mask $A \in \{0,1\}^{2L \times 2L}$ is then applied to control both causal and bidirectional connections.

This design allows simultaneous processing of corrupted and clean contexts, facilitating complementary mask supervision. The attention mask naturally supports both block-parallelism and causal autoregressive dependencies between blocks. We further employ the *flex-attention* implementation to efficiently realize this structured masking and accelerate training.

## 3.3 Inference Pipeline

At inference time, Fast-dLLM v2 employs a block-wise decoding strategy that balances the autoregressive nature of LLMs with the parallelism afforded by diffusion-based decoding. As shown in Figure 3, generation proceeds one block at a time: previously decoded blocks are cached and reused as clean prefix context, while the current block undergoes parallel masked token refinement.

By combining block-level caching, intra-block parallel decoding, and DualCache reuse, Fast-dLLM v2 maximizes decoding efficiency without requiring auxiliary models or extra inference overhead. Empirically, we observe up to $2.5\times$ speedup compared to standard AR decoding, while maintaining generation quality on par with strong autoregressive baselines. This makes Fast-dLLM v2 a compelling candidate for practical LLM deployment in latency-sensitive applications.

**Block-wise autoregressive decoding with caching.** Since each block in Fast-dLLM v2 is decoded in a causal order, we naturally preserve left-to-right semantics across blocks. After decoding each block, its unmasked tokens are cached as read-only context for future blocks. This design enables block-level Key-Value (KV) cache reuse and significantly reduces redundant computation. The attention mask at inference time allows each block to attend bidirectionally within itself, while attending causally to preceding blocks, mirroring the configuration used during training.

**Parallel refinement within each block.** To accelerate generation within a block, we adopt the confidence-aware parallel decoding strategy proposed in Fast-dLLM (Wu et al., 2025). Specifically, we iteratively refine masked tokens in the current block based on their model confidence of predicted tokens. Tokens exceeding a confidence threshold are decoded and unmasked in parallel, while uncertain positions remain masked for future refinement. This avoids ambiguous predictions and reduces generation latency.

**DualCache for sub-block reuse.** To further reduce redundant computation during intra-block decoding, we integrate the DualCache mechanism from Fast-dLLM. DualCache maintains both prefix and suffix KV caches for partially decoded blocks, enabling efficient recomputation as additional tokens are revealed. This hierarchical caching not only rules out expensive recomputation, but also supports the iterative, selective decoding pattern used in confidence-aware refinement.

**Batch decoding with padding.** To support batch generation of sequences with varying target lengths, we right-pad each sequence with `[MASK]` tokens to make their total lengths divisible by the block size $D$. The sequences are then grouped and decoded block-by-block. At each step, all sequences in the batch decode the next block in parallel, regardless of how many real tokens remain, ensuring consistent and efficient scheduling on modern hardware.

## 4 Experiments

### 4.1 Experimental Setup

We conduct adaptation experiments on the Qwen-2.5 1.5B and 7B Instruct models, tuning them into the Block Diffusion LLM configuration. For training, we use the LLaMA-Nemotron post-training dataset (Bercovich et al., 2025) with a batch size of 256. The learning rate and training steps are set specifically for each model: the 1.5B model is trained with a learning rate of $2 \times 10^{-5}$ for 6,000 steps, while the 7B model uses a learning rate of $1 \times 10^{-5}$ for 2,500 steps. We employ 64 NVIDIA A100 GPUs for training, with the 1.5B model training for approximately 8 hours and the 7B model for 12 hours. Unless otherwise stated, the sub-block size is fixed at 8, block size at 32, and parallel decoding is disabled (i.e., threshold is set to 1).

For benchmarking, we evaluate the tuned models on a comprehensive suite of tasks, covering diverse aspects of language modeling and reasoning abilities. The evaluation suite includes code generation tasks like HumanEval and MBPP, mathematical reasoning tasks such as GSM8K and MATH, as well as knowledge-intensive benchmarks like MMLU and GPQA, instruction-following tasks such as IFEval. Code-related benchmarks, including HumanEval and MBPP, are assessed using the EvalPlus framework, which provides a robust evaluation for code synthesis. All other benchmarks

Table 1: Benchmark results of various language models across a range of evaluation tasks. Models are grouped by parameter scale into 1B and 7B+ categories. Evaluation metrics include code generation (HumanEval, MBPP), mathematical reasoning (GSM8K, MATH), instruction following (IFEval), knowledge-intensive QA (MMLU, GPQA), and general average score (Avg.). "Base" and "Plus" refer to different evaluation settings for code benchmarks using EvalPlus. The best results per column are in **bold**, and the second-best are underlined.

| Model | #Params | HumanEval | | MBPP | | GSM8K | Math | IFEval | MMLU | GPQA | Avg. |
|---|---|---|---|---|---|---|---|---|---|---|---|
| | | Base | Plus | Base | Plus | | | | | | |
| **1B Models** | | | | | | | | | | | |
| LlaMA-3.2 | 1.2B | 34.1 | 31.1 | 34.1 | 29.4 | 43.0 | 23.8 | 58.9 | 44.4 | 24.1 | 35.9 |
| SmolLM 2 | 1.7B | 34.1 | 28.7 | 50.6 | 46.0 | 47.7 | 21.1 | 55.1 | 49.1 | 29.2 | 40.7 |
| Qwen2.5-1.5B | 1.5B | 42.1 | 37.2 | 48.1 | 41.3 | 57.0 | 46.8 | 41.2 | 54.6 | 30.6 | 44.3 |
| Qwen2.5-1.5B-Nemo-FT | 1.5B | 37.2 | 33.5 | 53.4 | 44.4 | 58.5 | 43.5 | 39.4 | 58.1 | 31.0 | 44.3 |
| **Fast-dLLM v2** | 1.5B | 43.9 | 40.2 | 50.0 | 41.3 | 62.0 | 38.1 | 47.0 | 55.1 | 27.7 | **45.0** |
| **7B+ Models** | | | | | | | | | | | |
| LLaDA | 8B | 35.4 | 31.7 | 31.5 | 28.6 | 78.6 | 26.6 | 59.9 | 65.5 | 31.8 | 43.3 |
| LLaDA-1.5 | 8B | 52.4 | - | 42.8 | - | 83.3 | 42.6 | 58.2 | 66.0 | 36.9 | - |
| LLaDA-MoE | 7B | 61.6 | - | 70.0 | - | 82.4 | 58.7 | 59.3 | 67.2 | - | - |
| Dream | 7B | 57.9 | 53.7 | 68.3 | 56.1 | 81.0 | 39.2 | 62.5 | 67.0 | 33.0 | 57.6 |
| Qwen2.5-7B | 7B | 51.2 | 47.6 | 57.7 | 49.5 | 71.4 | 73.3 | 70.8 | 68.7 | 33.5 | 58.2 |
| Qwen2.5-7B-Nemo-FT | 7B | 52.4 | 48.2 | 57.1 | 50.0 | 84.1 | 72.0 | 69.5 | 68.6 | 34.2 | 59.6 |
| **Fast-dLLM v2** | 7B | 63.4 | 58.5 | 63.0 | 52.3 | 83.7 | 61.6 | 61.4 | 66.6 | 31.9 | **60.3** |

are evaluated using the LM-Eval framework, ensuring consistency and reliability of performance measurements across different tasks.

To provide meaningful comparisons, we include several baseline models in our evaluation. These baselines include widely recognized models with comparable parameter sizes, such as LLaMA-3.2, SmolLM-2, Dream, and LLaDA series. Additionally, results from the original Qwen-2.5 1.5B and 7B models, tuned with next token prediction under the same dataset and training steps, are incorporated to highlight the impact of our Block Diffusion methodology.

## 4.2 MAIN RESULTS: PERFORMANCE AND SPEED

The 1.5B Fast-dLLM v2 achieves an average score of 45.0, outperforming the Qwen2.5-1.5B and Qwen2.5-1.5B-Nemo-FT baselines, and establishing new state-of-the-art performance among 1B-scale diffusion-based and NTP-trained autoregressive models. At the 7B scale, Fast-dLLM v2 reaches an average score of 60.3, surpassing all baselines including Qwen2.5-7B-Nemo-FT (59.6) and Dream (57.6), while matching or exceeding the best-performing models across most individual benchmarks. These results highlight Fast-dLLM v2's consistent and strong performance across diverse tasks, while maintaining the efficiency advantages of diffusion-based generation.

As shown in Table 1, our Fast-dLLM v2 models achieve competitive performance across a wide range of benchmarks. Notably, both the 1.5B and 7B variants perform on par with or better than their counterparts trained with standard next-token prediction loss on the same data and for the same number of steps.

To balance generation quality and efficiency, we adopt a confidence-based parallel decoding strategy, where each token is individually finalized once its predicted confidence exceeds a predefined threshold. As shown in Figure 4, a lower threshold allows more tokens to be finalized earlier in the denoising process, effectively reducing the number of required decoding steps and improving throughput. Specifically, with a threshold of 0.9, we observe only a marginal drop in GSM8K accuracy, while throughput increases

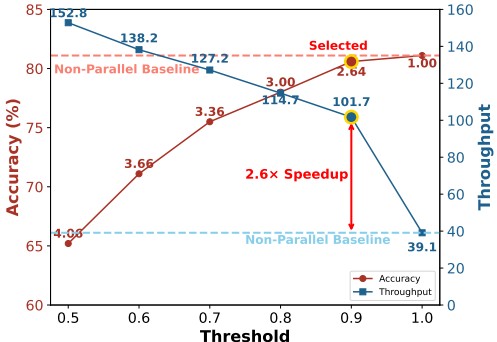

Figure 4: Accuracy and throughput under different thresholds on GSM8K. Threshold 0.9 is selected, offering a 2.6× speedup with minimal accuracy drop.

significantly from 39.1 to 101.7 tokens/s—yielding a 2.6× speedup. This setting provides a favorable trade-off between performance and efficiency. Importantly, setting the threshold to 1.0 recovers the standard non-parallel decoding process, where all tokens are updated through the full sequence of denoising steps and finalized only at the end.

Figure 5 compares the throughput of Fast-dLLM v2 (7B) and Qwen2.5-7B-Instruct across a range of batch sizes on both NVIDIA A100 and H100 GPUs for GSM8K, where we set the threshold to 0.9 and use sub-block cache. Across all settings, diffusion generation consistently outperforms the autoregressive baseline, demonstrating superior scalability with increasing batch size. On the A100, diffusion achieves up to 1.5× higher throughput at batch size 64, while the advantage is even more pronounced on the H100, reaching up to 1.8× speedup. This improvement highlights the efficiency benefits of diffusion decoding, especially on newer hardware architectures where parallelism can be better exploited. These results reinforce the practicality of diffusion-based generation in real-world deployment scenarios where low-latency, high-throughput inference is critical.

### 4.3    ABLATION STUDY

We conduct all ablation experiments using the Fast-dLLM v2 1.5B model to systematically investigate the impact of architectural and decoding choices. As shown in Table 2, the baseline ("naive token shift") applies a strategy where, for each training block, a subset of tokens is randomly masked, and each masked token is predicted using the model's output at the preceding position. To improve training fidelity, we introduce a padding strategy ("+ pad") that appends non-loss-bearing <MASK> tokens to each training sample such that its length becomes a multiple of block size. This modification is crucial to preserve data integrity during sequence packing: without padding, an <EOS> token from one sample might be immediately followed by a <BOS> token from the next, and since our block-wise diffusion model uses bidirectional attention, this can lead to unintended attention across samples. Padding ensures clean sample boundaries, preventing cross-sample leakage during training.

We further incorporate a complementary masking (CM) strategy, where the complement of each sampled mask is also used in training. This ensures that all tokens in the input receive supervision, increasing the coverage of the learning signal. The full recipe ("+ pad + CM") achieves the best overall performance across benchmarks, improving the average accuracy by +3.7 points over the naive strategy. These results highlight the importance of aligning training-time input construction with the model's attention mechanism and masking objectives.

As shown in Table 3 and Table 4, we explore how sub-block size and block size affect final performance. In Table 3, we observe that adjusting the *sub-block size* during inference leads to notable gains, with a size of 8 achieving the highest accuracy on average across tasks. While GSM8K performs best with smaller sizes (e.g., 2), HumanEval and HumanEval+ show improved results up to size 8, indicating that the optimal sub-block size is task-dependent.

In contrast, Table 4 illustrates that directly modifying the *block size* at inference time—without aligning it to the training-time configuration—results in substantial performance degradation. For instance, GSM8K performance drops from 62.0 in the sub-block setting to 58.5 under mismatched block size, and HumanEval shows similar trends. This disparity underscores the importance of maintaining consistency between training and inference block

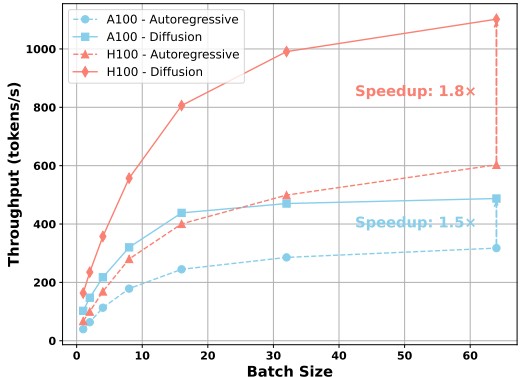

Figure 5: Throughput comparison between autoregressive and diffusion generation methods on NVIDIA A100 and H100 GPUs across varying batch sizes. Diffusion generation consistently outperforms autoregressive on both GPUs.

structures. By introducing a sub-block decoding strategy, we are able to flexibly control inference granularity without violating this consistency, thereby achieving better performance across diverse benchmarks.

Table 2: Benchmark results for different token shift strategies. "+ CM" stands for "+ complementary mask". The best performance for each benchmark is shown in **bold**, while the second-best is underlined.

| Method | HumanEval | | MBPP | | GSM8K | Math | IFEval | MMLU | GPQA | Avg. |
|---|---|---|---|---|---|---|---|---|---|---|
| | Base | Plus | Base | Plus | | | | | | |
| Naive token shift | 38.4 | 32.9 | 44.4 | 38.6 | 59.0 | 37.3 | 39.9 | 52.9 | **27.9** | 41.3 |
| + pad | 38.4 | 34.1 | 45.2 | 38.4 | 60.1 | 37.0 | 45.8 | 53.5 | 27.7 | 42.2 |
| + pad + CM | **43.9** | **40.2** | **50.0** | **41.3** | **62.0** | **38.1** | **47.0** | **55.1** | 27.7 | **45.0** |

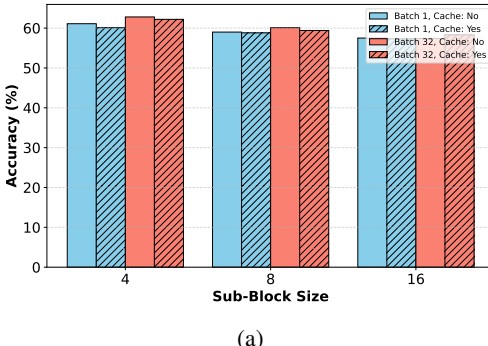

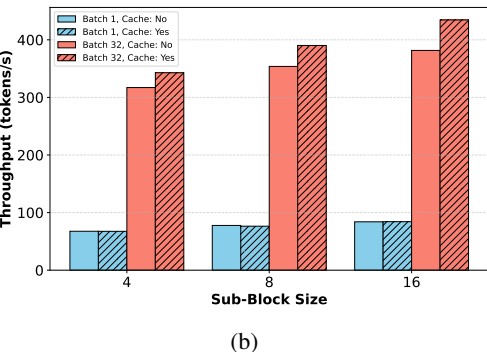

(a)                                    (b)

Figure 6: Effect of small block size and sub-block cache on model performance. (**a**) Accuracy remains largely unaffected by the use of sub-block cache across different block sizes and batch sizes. (**b**) Throughput increases as small block size grows due to higher decoding parallelism. While sub-block cache has negligible effect when batch size is small, it significantly improves throughput under compute-bound settings (e.g., batch size = 32).

We further study the effects of sub-block size and sub-block cache on both accuracy and throughput, as illustrated in Figure 6. In Figure 6a, we observe that increasing the sub-block size leads to a slight drop in accuracy, consistent with previous findings in Table 3. As shown in Figure 6b, using larger sub-block sizes increases decoding throughput by decreasing the number of required sequential forward passes and exploiting greater intra-step parallelism.

In addition, we evaluate the impact of introducing a sub-block cache. While the cache introduces negligible gains when the batch size is small (and memory bandwidth is underutilized), it provides substantial speedup in the compute-bound regime, such as when the batch size is 32 as shown in Figure 5. Importantly, caching has no observable effect on model accuracy (Figure 6a), confirming that it is a purely efficiency-enhancing feature without compromising output quality. These results highlight that under practical batch sizes, combining larger sub-block sizes with caching yields strong performance-efficiency benefits.

Table 3: Sub-Block size decoding improves performance, with size 8 being optimal.

| Sub-Block Size | 2 | 4 | 8 | 16 | 32 |
|---|---|---|---|---|---|
| GSM8K | **62.8** | 61.8 | 62.0 | 61.3 | 60.2 |
| HumanEval | 42.7 | 43.3 | **43.9** | 39.6 | 38.4 |
| HumanEval+ | 39.6 | **40.2** | **40.2** | 36.0 | 34.8 |

Table 4: Inference with mismatched sizes reduces performance.

| Block Size | 2 | 4 | 8 | 16 | 32 |
|---|---|---|---|---|---|
| GSM8K | 53.2 | 56.8 | 58.5 | 59.7 | **60.2** |
| HumanEval | 37.8 | **43.3** | **43.3** | 38.4 | 38.4 |
| HumanEval+ | 34.1 | 39.0 | **39.6** | 34.1 | 34.8 |

## 5 CONCLUSION

In this work, we presented Fast-dLLM v2, a scalable block diffusion language model framework that adapts pretrained autoregressive LLMs into efficient diffusion-style decoders for parallel text gener-

ation. By integrating a blockwise diffusion mechanism with complementary masking, Fast-dLLM v2 enables intra-block bidirectional context modeling while retaining the predictive capabilities of the original AR models. To address the latency of existing diffusion-based models, we further proposed a hierarchical caching strategy, consisting of a block-level cache for inter-block context reuse and a DualCache-based sub-block cache for efficient refinement within blocks, together with a parallel decoding pipeline. Extensive experiments on large-scale Qwen2.5-Instruct models (1.5B and 7B) demonstrate that Fast-dLLM v2 achieves up to $2.5\times$ speedup over standard AR decoding without loss of generation quality, consistently matching strong AR baselines while surpassing prior diffusion-based approaches in efficiency. These results highlight the potential of block diffusion frameworks as a practical path toward deploying high-quality, low-latency LLMs in real-world applications.

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

# A    IMPLEMENTATION DETAILS

## A.1    TRAINING SETUP

We fine-tune pretrained Qwen2.5-Instruct models (1.5B and 7B) under our block-wise diffusion training framework. Unless otherwise specified, all experiments adopt a context length of 2048 and batch size of 256. Training is conducted on 64 NVIDIA A100 GPUs using DeepSpeed Zero-3.

**Training Data.**    Our models are fine-tuned on a subset of the LLaMA-Nemotron post-training dataset, which contains high-quality instruction-following examples covering a broad range of domains. We preprocess the dataset using block-wise packing, and pad each sequence to a multiple of the block size to avoid misaligned block boundaries. Redundant padding tokens are excluded from loss computation and gradient updates.

**Hyperparameters.**    The 1.5B model is trained for 6,000 steps with a learning rate of $2 \times 10^{-5}$, while the 7B model is trained for 2,500 steps with a learning rate of $1 \times 10^{-5}$. In both settings, we use AdamW as the optimizer and apply linear learning rate warmup over the first 500 steps. With a context length of 2048 and batch size of 256, each training step processes $256 \times 2048 = 524{,}288$ tokens. This corresponds to a total training token count of approximately:

- 1.5B model: $6{,}000 \times 524{,}288 \approx 3.15$ billion tokens
- 7B model: $2{,}500 \times 524{,}288 \approx 1.31$ billion tokens

We fix the block size to 32 for all experiments. All training sequences are right-padded and packed in a block-aligned fashion to fully utilize model context, enabling consistent and efficient batch construction under hardware constraints.

## A.2    ATTENTION MASK DESIGN

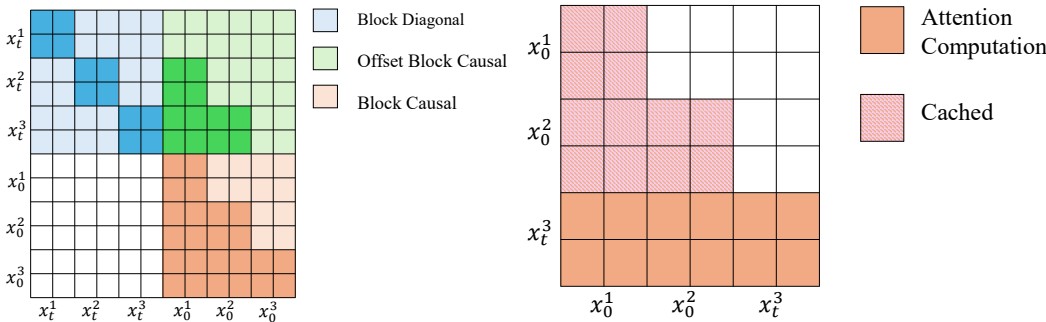

(a) Training-time attention mask.        (b) Inference-time attention mask.

Figure 7: Specialized attention mask design for diffusion language modeling. **(a)** During training, each input consists of a corrupted sequence $x_t$ and corresponding targets $x_0$, concatenated and processed in a single forward pass. The attention mask combines intra-block bidirectional attention (Block Diagonal), cross-block causal dependency from clean tokens to noised ones (Offset Block Causal), and traditional left-to-right causality among clean tokens (Block Causal). **(b)** During inference, previously decoded blocks of $x_0$ are reused via caching. Only the current noised block $x_t$ is computed in each decoding step, which attends to cached prefixes (shaded) and updates its own block in a self-contained fashion.

To enable efficient and structured learning across both corrupted and clean views of the input, we use a custom block-aware attention scheme (Arriola et al., 2025). At each training step, we concatenate the noised sequence $x_t$ and the clean sequence $x_0$ into a single input of total length $2L$, then apply a hybrid attention pattern defined via an attention mask $\mathcal{M}_{\text{full}} \in \{0, 1\}^{2L \times 2L}$.

To simplify notation, we follow prior work and slightly abuse the symbol $x^b$, which in this context denotes the set of tokens in the $b$-th block (rather than the $b$-th token, as in earlier sections). Specifically, we aim to model the conditional probabilities $p_\theta(x^b \mid x_t^b, x^{<b})$ across all blocks $b \in [1, B]$, where $x_t^b$ is the noised version of block $b$, and $x^{<b}$ comprises all clean tokens in previous blocks. This formulation enables us to process both noised and clean representations simultaneously by feeding their concatenated sequence into the transformer and applying a carefully constructed attention mask $\mathcal{M}_{\text{full}}$ as shown in Figure 7(a).

The overall attention mask can be decomposed into four sub-masks:

$$\mathcal{M}_{\text{full}} = \begin{bmatrix} \mathcal{M}_{BD} & \mathcal{M}_{OBC} \\ 0 & \mathcal{M}_{BC} \end{bmatrix},$$

where:

- $\mathcal{M}_{BD}$ (Block-diagonal mask): Provides bidirectional self-attention among tokens within the same block in the noised sequence $x_t$, enabling within-block refinement:

$$[\mathcal{M}_{BD}]_{ij} = \begin{cases} 1 & \text{if } i, j \text{ belong to the same block} \\ 0 & \text{otherwise} \end{cases}$$

- $\mathcal{M}_{OBC}$ (Offset block-causal mask): Allows each noised token in $x_t$ to attend to tokens from previous blocks in the clean sequence $x_0$, preserving inter-block causal conditioning:

$$[\mathcal{M}_{OBC}]_{ij} = \begin{cases} 1 & \text{if } j \text{ is in a block before } i \\ 0 & \text{otherwise} \end{cases}$$

- $\mathcal{M}_{BC}$ (Block-causal mask): Enables each token in the clean sequence $x_0$ to attend to all previous and current block positions, facilitating autoregressive-like progression:

$$[\mathcal{M}_{BC}]_{ij} = \begin{cases} 1 & \text{if } j \text{ is in the same or an earlier block as } i \\ 0 & \text{otherwise} \end{cases}$$

The combined mask allows unified handling of masked token prediction, simultaneous conditioning on prior known context, and structural training efficiency via block-parallelism.

During inference, we adopt a simplified causal attention mechanism that reuses decoded blocks as frozen prefix context. As illustrated in Figure 7(b), previously generated blocks from $x_0^{<b}$ are cached to avoid redundant computation, and only the current noised block $x_t^b$ is actively refined. This block attends bidirectionally within itself, similar to $\mathcal{M}_{BD}$ during training, while attending causally to the unmasked tokens in previous blocks. The attention computation is thus restricted to the current block and its causal prefix, enabling efficient decoding via key-value cache reuse and reduced memory footprint. This structure preserves left-to-right semantics across blocks while allowing intra-block denoising in parallel.

## A.3 DETAILS ON TRAINING OBJECTIVE

We minimize the masked-token-only cross-entropy loss:

$$\mathcal{L}_{\text{block}}(\theta) = -\mathbb{E}_{x,m} \left[ \sum_{i=1}^{L} \mathbf{1}[x_t^i = \text{[MASK]}] \log p_\theta(x_0^i \mid x_{<i}, x_{\text{block}(i)}) \right].$$

Notably, this objective function seems to omit the normalization coefficient $\frac{1}{t}$ often found in standard masked modeling losses (e.g., dividing by the number of masked tokens). This is intentional and justified by our complementary masking strategy. This is because we use a complementary mask for each training sample $x_0$: we always sample two complementary times $t$ and $1 - t$ with mask $m$ and $\bar{m} = 1 - m$.

$$-\left[ \sum_{i=1}^{L} \mathbf{1}[x_t^i = \text{[MASK]}] \log p_\theta(x_0^i \mid x_{<i}, x_{\text{block}(i)}) \right] + \left[ \sum_{i=1}^{L} \mathbf{1}[x_{1-t}^i = \text{[MASK]}] \log p_\theta(x_0^i \mid x_{<i}, x_{\text{block}(i)}) \right].$$

Due to the complementary mask, the total number of tokens contributing to the loss for any given sample $x_0$ is always the full sequence length $L$.

## A.4 EVALUATION PROTOCOL

We evaluate all trained models on a diverse suite of downstream benchmarks covering reasoning, knowledge, and code generation. Unless otherwise specified, all evaluations are conducted using greedy decoding (`argmax`). We adopt zero-shot settings for all tasks, with the exception of GPQA, which is evaluated under 5-shot prompting following standard protocol.

All non-code tasks are evaluated using the LM-Eval harness, ensuring compatibility and fair performance reporting. For code tasks like HumanEval and MBPP, we employ the EvalPlus framework for reliable pass-rate calculation. Unless otherwise noted, the following setup is used during inference:

- Block size = 32
- Sub-block size = 8
- Parallel decoding disabled (threshold = 1)

This configuration ensures consistency between training and inference setups, facilitating effective evaluation of the block-wise diffusion capability in Fast-dLLM v2.

## B CASE STUDY

To better illustrate the reasoning and interaction capabilities of Fast-dLLM v2 (7B), we conducted a detailed examination of both single-turn and multi-turn dialogue scenarios. Representative examples are presented in Table 5 and Table 6.

**Single-turn Dialogue Scenarios.** As shown in Table 5, Fast-dLLM v2 is capable of handling complex queries in a single interaction. In the *Math* example, the model correctly analyzes the rational function to determine the number of vertical asymptotes, applying algebraic factoring and solving for the undefined values of $x$. In the *Code* section, the model generates a correct and recursive Python implementation of the Tower of Hanoi problem, along with an appropriate explanation of the input parameters and output.

**Multi-turn Dialogue Scenarios.** Table 6 highlights multi-turn dialogues where Fast-dLLM v2 retains context and builds upon previous turns. The *Daily life* example illustrates the model's ability to perform temporal reasoning, such as computing the number of years since a company was founded and determining its future anniversary. The *Math* example showcases step-by-step logical reasoning to solve a real-world arithmetic problem involving truckload capacity and total delivery time. The model effectively breaks the problem into sequential steps, performs intermediate calculations, and presents the final result in the required format (hours and minutes).

These case studies collectively demonstrate Fast-dLLM v2's strength in mathematical reasoning, code generation, temporal understanding, and contextual coherence across both single-turn and multi-turn settings.

## C ADDITIONAL EXPERIMENTS

### C.1 LATENCY AND THROUGHPUT SCALING WITH CONTEXT LENGTH

To evaluate the scalability of our method in deployment scenarios involving longer inputs, we extended our testing beyond the sequence lengths reported in the main text. We conducted experiments varying the context length from 1K to 8K tokens, measuring both generation latency and throughput for generating 32 completion tokens. All tests were performed on a single NVIDIA A100 80GB GPU.

As illustrated in Figure 8, our hierarchical cache consistently outperforms block-only cache, particularly at longer contexts. Notably, it also achieves latency and throughput comparable to standard AR model caching, showing that our caching system is highly effective in reducing overhead even under blockwise diffusion decoding. Furthermore, when combined with our parallel decoding strategy (i.e., sub-block speculative denoising), the overall method achieves 2–3× speedup over strong AR baselines — as shown in Figure 1 of the main paper.

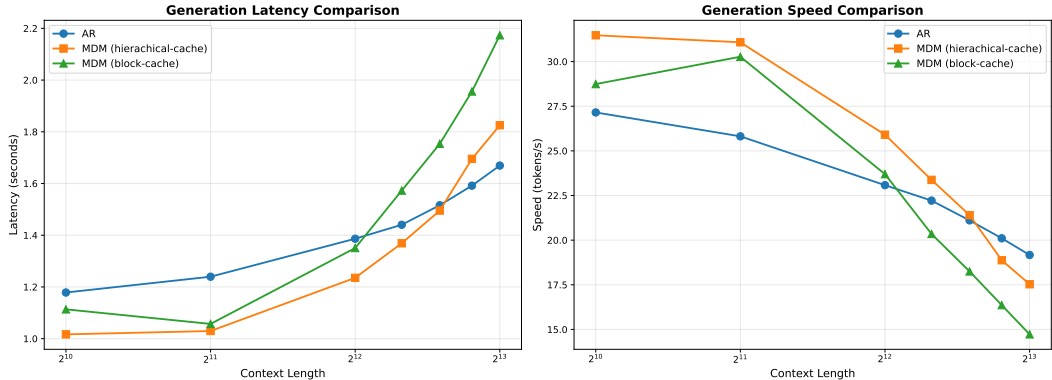

Figure 8: **Latency and Throughput Scaling.** Comparison of generation latency and throughput across context lengths ranging from 1K to 8K tokens. Our hierarchical cache maintains high efficiency as context length increases.

# D LLM USAGE

During manuscript preparation, we used large language models —strictly for language polishing of paragraphs and sentences (grammar, flow, and tone). These tools were not used to generate ideas, design experiments, or determine conclusions. All technical content, methodology, and interpretations were written, verified, and approved by the authors. To reduce risks of factual drift or citation errors, we required human review of every model-edited sentence and cross-checked all references against primary sources. The authors take full responsibility for the accuracy and integrity of the manuscript.

Table 5: **Single-turn Dialogue Cases of Fast-dLLM v2 (7B).**

*Math*

**User**    How many vertical asymptotes does the graph of $y = \frac{2}{x^2+x-6}$ have?

**Assistant**    To determine the number of vertical asymptotes of the function $y = \frac{2}{x^2+x-6}$, we need to identify the values of $x$ that make the denominator zero, as these values will cause the function to approach infinity, indicating a vertical asymptote.

First, we factor the denominator $x^2 + x - 6$. We look for two numbers that multiply to $-6$ and add up to $1$. These numbers are $3$ and $-2$. Therefore, we can factor the quadratic as follows:

$$x^2 + x - 6 = (x + 3)(x - 2)$$

So the function can be rewritten as:

$$y = \frac{2}{(x + 3)(x - 2)}$$

The denominator is zero when $x + 3 = 0$ or $x - 2 = 0$. Solving these equations, we get:

$$x = -3 \quad \text{and} \quad x = 2$$

Since the function is undefined at $x = -3$ and $x = 2$, there are vertical asymptotes at these points. Therefore, the function $y = \frac{2}{x^2+x-6}$ has two vertical asymptotes.

The number of vertical asymptotes is $\boxed{2}$

*Code*

**User**    Write a recursive Python function to solve the Tower of Hanoi problem and print each move.

**Assistant**

```python
def tower_of_hanoi(n, source, target, auxiliary):
    if n > 0:
        tower_of_hanoi(n - 1, source, auxiliary, target)
        print(f"Move disk {n} from {source} to {target}")
        tower_of_hanoi(n - 1, auxiliary, target, source)

tower_of_hanoi(3, 'A', 'C', 'B')
```

This function solves the Tower of Hanoi problem using recursion. It takes four parameters: `n` (the number of disks), `source` (the starting peg), `target` (the destination peg), and `auxiliary` (the helper peg). The function prints each move in the form of "Move disk n from source to target".

Table 6: **Multi-turn Dialogue Cases of Fast-dLLM v2 (7B).**

*Daily life*

| | |
|---|---|
| **User** | When was NVIDIA founded? |
| **Assistant** | NVIDIA was founded in 1993. |
| **User** | Using the year you just gave me, calculate how many years NVIDIA has been in business as of 2025. |
| **Assistant** | NVIDIA has been in business for 32 years as of 2025. |
| **User** | How many years will it be until NVIDIA's 40th anniversary? |
| **Assistant** | NVIDIA's 40th anniversary will be in 2033, which is 8 years from now. |

*Math*

**User** A delivery truck can carry 120 boxes in one trip. If a warehouse needs to send out 960 boxes, how many trips will the truck need?

**Assistant** To determine how many trips the delivery truck will need to send out 960 boxes, we start by noting the capacity of the truck in one trip. The truck can carry 120 boxes in one trip. We need to find out how many trips are required to transport 960 boxes.
We can set up the problem as a division problem:

$$\text{Number of trips} = \frac{\text{Total number of boxes}}{\text{Capacity of the truck per trip}}$$

Substituting the given values into the equation, we get:

$$\text{Number of trips} = \frac{960}{120}$$

Next, we perform the division:

$$\frac{960}{120} = 8$$

Therefore, the truck will need $\boxed{8}$ trips to send out 960 boxes.

**User** If each trip takes 45 minutes, how long will it take to deliver all boxes in total? Give your answer in hours and minutes.

**Assistant** To determine the total time required to deliver all the boxes, we need to follow these steps:
1. Calculate the number of trips required.
2. Determine the total time taken for all the trips.
3. Convert the total time from minutes to hours and minutes.
**Step 1: Calculate the number of trips required.**
The warehouse needs to send out 960 boxes, and the truck can carry 120 boxes per trip. Therefore, the number of trips required is:

$$\text{Number of trips} = \frac{960}{120} = 8$$

**Step 2: Determine the total time taken for all the trips.**
Each trip takes 45 minutes, so the total time taken for 8 trips is:

$$\text{Total time} = 8 \times 45 = 360 \text{ minutes}$$

**Step 3: Convert the total time from minutes to hours and minutes.**
There are 60 minutes in an hour, so we convert 360 minutes to hours by dividing by 60:

$$360 \div 60 = 6 \text{ hours}$$

Therefore, the total time required to deliver all the boxes is:

$$\boxed{6 \text{ hours}}$$

