# OpenReview forum: "Fast-dLLM v2: Efficient Block-Diffusion LLM"
_ICLR.cc/2026/Conference — ICLR 2026 Poster_

### Official Review · Reviewer_aB3q · 2025-10-31

**Soundness:** 4
**Presentation:** 3
**Contribution:** 4
**Rating:** 8
**Confidence:** 4

**Summary:**

This paper introduces Fast-dLLM v2, a block diffusion language model designed to address the inherent inefficiency of sequential decoding in standard autoregressive LLMs. Its core contribution is a highly data-efficient adaptation method that converts pre-trained AR models into efficient parallel decoders, requiring around ~1B tokens for the entire fine-tuning process. This represents a 500x reduction in required training data compared to full-attention dLLMs such as Dream, while successfully preserving the original model's performance. The method employs a novel training recipe combining a block diffusion mechanism with a complementary attention mask to enable block-wise bidirectional context modeling. To accelerate inference, it introduces a hierarchical caching mechanism: a "block-level cache" for inter-block context reuse, and a DualCache-based "sub-block cache" to facilitate efficient parallel refinement within each block. Extensive experiments demonstrate that Fast-dLLM v2 achieves up to a 2.5x inference speedup compared to standard AR decoding. Crucially, this acceleration is achieved without sacrificing generation quality; the model's performance generally matches or exceeds that of strong AR baselines, while varying among different tasks.

**Strengths:**

1.The paper proposes a novel and ingenious method. The paper's design of the hierarchical caching mechanism is ingenious, maximizing the dLLM's intra-block parallel decoding potential while preserving the AR model's inter-block causal dependencies.

2.The paper's results are outstanding. The proposed training method is highly efficient and achieves SOTA inference speed without sacrificing quality. This greatly reduces training costs and holds significant practical value.

3.The experimental analysis is thorough and detailed. For instance, it provides a practical trade-off analysis of the "confidence threshold," which is highly instructive for the technology's real-world deployment.

4.The technical details and experimental settings are clearly articulated, facilitating reproducibility.

5.The paper's writing and organizational structure are very clear, demonstrating excellent readability.

Overall, this is a solid and good paper.

**Weaknesses:**

The paper claims that token shift strategy is necessary for preserving the pretrained AR model’s representation quality. However, this claim is not empirically validated with a dedicated ablation study. As shown in Table 2, the most basic baseline, 'naive token shift,' already includes this strategy. Therefore, the actual necessity of this design is unclear.

**Questions:**

1.The proposed method includes a hierarchical caching mechanism and a specialized hybrid attention mask; will this lead to a more complex engineering implementation?

2.The adaptation is performed using the LLaMA-Nemotron dataset, which consists of high-quality SFT data. This raises the question of whether such high-quality instruction data is a necessary component for successfully learning the block diffusion mechanism, or if the method could be equally effective using general pre-training corpora.

3.Could the same impressive results be achieved if other mainstream AR architectures were used for the pre-trained initialization? Or is the success of this method dependent on the specific architecture or quality of the AR model used?

---

> ### Author Response · Authors · 2025-11-22
> **Response to Reviewer aB3q**
>
> ## Q1: Design for token shift
> ## A1:
> Thank you for raising this thoughtful point. Our adoption of token shift is based on strong prior evidence from several recent diffusion-based LLM works: DiffuLLaMA (arXiv:2410.17891), FUDOKI (arXiv:2505.20147), and Dream (arXiv:2508.15487). These works consistently demonstrate that token shift is an effective strategy for aligning the diffusion denoising objective with the autoregressive generation behavior inherited from pretrained LMs. For this reason, we treat token shift as part of our default training setup — not as a novel contribution, but as a standard, empirically supported component. Our goal is to study how pretrained AR models can be adapted efficiently for blockwise diffusion decoding, and we used the best-known practices (including token shift) to ensure high adaptation quality.
>
> ## Q2: Implementation of hierarchical cache
> ## A2:
> While our method introduces new components — such as hierarchical caching and hybrid blockwise attention masks — we have designed them to be modular and engineering-friendly.
> Specifically:
> - The hierarchical cache builds directly on existing KV cache abstractions and adds a clean separation between inter-block and intra-block reuse.
> - The hybrid attention mask (causal across blocks, bidirectional within sub-blocks) is computed efficiently and integrates smoothly into standard attention implementations.
>
> Importantly, we are currently working to integrate our method into open-source inference frameworks like vLLM and SGLang. We are determined to make our system is compatible with real-world, high-performance serving pipelines. We expect that once integrated, the added complexity will be minimal compared to the practical gains in throughput and flexibility.
>
> ## Q3: Does block diffusion require high-quality instruction data or can it use general corpora?
> ## A3:
> Thank you for the thoughtful question. We clarify that the use of high-quality SFT data (LLaMA-Nemotron) in our work is a strategic choice to support efficient adaptation, not a requirement for the block diffusion mechanism itself. Our goal is to enable low-cost adaptation of pretrained AR models using only ~1B tokens — hence we used strongly supervised data to make the most of this limited budget. However, the blockwise formulation itself is data-agnostic. In fact, concurrent work Efficient-DLM (https://openreview.net/pdf?id=kIbFgoCq86) shows that block-based AR-to-diffusion adaptation can be trained effectively with general pretraining corpora (10B–100B tokens). This validates that block diffusion works well even without instruction tuning.
>
> In short, our work focuses on the final-stage adaptation setting, while Efficient-DLM demonstrates the same mechanism in continued pretraining. Both show that block diffusion is robust across different training regimes and data sources.
>
> ## Q4: Other mainstream AR architecture experiments
> ## A4:
> Thank you for raising this point — we have explicitly tested our method using another mainstream AR architecture, LLaMA3.2-1B, to verify generality.
> As shown in the table below, Fast-dLLM applied to LLaMA3.2-1B achieves comparable GSM8K accuracy to the original AR model (43.0 vs. 43.4) while delivering substantial throughput gains.
>
> The results show that even when starting from a different AR backbone, our adaptation framework maintains strong accuracy and yields up to ~1.8× speedup (with threshold=0.9), testing with A100-80G GPU, batch size=1.
>
> This confirms that the success of our method is not tied to a specific architecture — it is a general approach that can be applied to various pretrained AR models to achieve efficient decoding.
>
> |                   | LLaMA3.2-1B | Fast-dLLM-LLaMA3.2-1B (threshold 1) | Fast-dLLM-LLaMA3.2-1B (threshold 0.9) |
> |-------------------|-------------|-----------------------------------------|------------------------------------------|
> | GSM8K Acc         | 43.0        | 43.4                                    | 43.4                                     |
> | GSM8K Throughput        | 70.9        | 63.3                                    | 127.1                                    |

---

### Official Review · Reviewer_9tWZ · 2025-10-31

**Soundness:** 3
**Presentation:** 3
**Contribution:** 2
**Rating:** 4
**Confidence:** 3

**Summary:**

This paper proposes Fast-dLLM v2, a block diffusion language model (dLLM) designed for efficient inference. The core idea is to adapt a pretrained autoregressive (AR) LLM into a block-wise diffusion model using a data-efficient finetuning process (requiring only ~1B tokens). The method introduces a novel training recipe with a complementary attention mask to enable this adaptation. For inference acceleration, the paper proposes a hierarchical caching mechanism: a block-level cache for reusing context from previous blocks and a sub-block cache (adopting DualCache from Fast-dLLM v1) for parallel generation within the current block. Experiments show that Fast-dLLM v2 can achieve up to a 2.5x speedup over its AR baseline (Qwen2.5-7B) while maintaining comparable or slightly improved accuracy on several benchmarks.

**Strengths:**

1. Significant Problem: The paper addresses the critical problem of LLM inference latency. The goal of achieving parallel decoding speedups while preserving the quality of strong AR models is highly relevant.

2. Strong Empirical Results: The method demonstrates impressive practical results. A 2.5x speedup over a strong AR baseline like Qwen2.5-7B-Instruct with no loss in generation quality is a significant engineering achievement.

3. Data and Training Efficiency: A major strength is the data efficiency of the finetuning process. Adapting a 7B model with only ~1B tokens is a practical and accessible approach compared to training full-attention dLLMs, which the paper notes can require orders of magnitude more data (e.g., 580B tokens for Dream).

4. Clarity: The paper is well-written, and the proposed architecture, training process, and inference pipeline are explained clearly.

**Weaknesses:**

The primary weakness of this paper is its perceived lack of conceptually new contribution, as it appears to be a skillful integration of several existing lines of work.

1. Combination of Existing Caching Techniques: The "hierarchical caching mechanism" seems to be a straightforward combination of two existing ideas.
- The block-level cache, which stores representations of previously decoded blocks, is an inherent feature of the block-wise autoregressive structure, as seen in "Block Diffusion" (Arriola et al., 2025).
- The sub-block cache for parallel refinement within a block is explicitly stated to "adopt the DualCache in Fast-dLLM (Wu et al., 2025)" (Section 3.3).
- While effective, combining these two caches (inter-block from Block Diffusion, intra-block from Fast-dLLM v1) feels more like an incremental engineering step than a novel caching design.

2. Finetuning Concept is Not New: The core idea of adapting a pretrained AR model into a diffusion model has been explored before.

- Previous work Dream (Ye et al., 2025b;a) is already a model that was "adapted from the existing Qwen-2.5 7B" (Section 2.1), and the paper also points to "SDAR (Cheng et al., 2025)" as concurrent work that "successfully finetuned a block diffusion model from a pretrained autoregressive model" (Section 2.2).
- The main claim over Dream is data efficiency (1B vs. 580B tokens). However, this comparison may not be direct, as Dream adapts to a full-attention dLLM, whereas this work adapts to a more structured block-wise dLLM. The efficiency gain may stem more from the architectural choice (block-wise vs. full-attention) than from a fundamentally new finetuning recipe.

Overall, while the final system is fast and effective, the contributions seem to be more in the successful engineering and combination of existing components rather than the introduction of new, fundamental concepts.

**Questions:**

1. Could the authors please elaborate on the novelty of the hierarchical caching mechanism? Is there more to it than the combination of the inter-block cache from Block Diffusion and the intra-block DualCache from Fast-dLLM v1?

2. The data efficiency (1B vs. 580B tokens) is a key claim over Dream. How much of this 500x reduction is due to the proposed training recipe (complementary masks, etc.) versus the simpler, more AR-like target architecture (block-wise dLLM) compared to Dream's full-attention dLLM?

3. Could the authors clarify the primary novel contributions of their finetuning approach and inference pipeline compared to the concurrent SDAR (Cheng et al., 2025), which also finetunes a pretrained AR model into a block diffusion model?

---

> ### Author Response · Authors · 2025-11-22
> **Response to Reviewer 9tWZ (1/2)**
>
> ## Q1: Novelty of hierarchical cache
> ## A1:
> While it is true that our hierarchical caching mechanism draws on ideas from prior work — such as block-level caching (from Block Diffusion) and DualCache (from Fast-dLLM v1) — we respectfully argue that Fast-dLLM v2 goes beyond a simple combination, and introduces a unified, scalable, and high-performance system for diffusion-based LLM generation.
> 1. Block-diffusion: Although Block Diffusion does use block-level caching, it is limited to small-scale models (400M) and does not deliver system-level acceleration or support fine-grained control over decoding. In contrast, our hierarchical cache is carefully co-designed with sub-block-aware decoding to reduce both computation and memory bottlenecks. As shown in Figures 1, 5 and 6, our design achieves significant speedups over autoregressive baselines while maintaining comparable performace — something prior diffusion models did not realize.
> 2. Sub-Block Decoding: Neither Block Diffusion nor Fast-dLLM v1 design mechanisms to address block size mismatch between training and inference — this is a critical limitation in practical deployment. Sub-block decoding in our method solves this by allowing fine-grained intra-block progression, enabling efficient partial caching and decoding without retraining. As shown in Tables 3 and 4, this results in higher accuracy and more flexible generation compared to variants without sub-blocks.
> 3. More than a Mechanical Combination: Designing a hierarchical cache that coherently unifies inter-block and intra-block reuse — in the presence of hybrid attention masks, dynamic decoding schedules, and autoregressive generation constraints — required new algorithmic and engineering solutions. Our method is validated not just for feasibility, but for practical LLM-scale effectiveness.
>
> In summary, while we acknowledge and build upon valuable prior work, Fast-dLLM v2 brings these ideas into a unified, scalable, and production-ready architecture.
>
> ## Q2: Finetuning Concept is Not New: The core idea of adapting a pretrained AR model into a diffusion model has been explored before
> ## A2:
>
> We agree that the idea of adapting a pretrained AR model to a diffusion model is not entirely new — prior and concurrent works such as Dream and SDAR also explore this direction. However, we would like to clarify several key distinctions in our work:
> * SDAR is a concurrent work, independently developed and released around the same time as ours. While it supports the general idea of AR-to-diffusion adaptation, it should not be viewed as prior art relative to our method.
> * Our contribution lies in building a systematic and highly efficient adaptation framework, specifically:
>   - We target blockwise diffusion decoding, which introduces challenges absent in full-attention diffusion (e.g., training–inference block size mismatch).
>   - To tackle this, we introduce sub-block decoding, a novel decoding strategy that allows autoregressive, left-to-right sub-block generation within a block, improving both robustness and flexibility. This is validated in Table 3 vs. Table 4, where we show significant performance degradation under naive block-mismatch inference.
> * Our training framework incorporates several unique components that further enable high data efficiency:
>   * Complementary masking improves denoising signal within small blocks (Table 2: +2.8 average improvement).
>   * Smart padding schemes prevent unnecessary information distortion in short sequences for sequence-packing.
> * While Dream uses 580B tokens for full-attention diffusion retraining, we demonstrate that our method can reach comparable — and in many cases stronger — performance using just 1B tokens. As shown in Table 1, Fast-dLLM v2 outperforms Dream across nearly all tasks, and achieves average performance comparable to SoTA AR baselines (e.g., Qwen-2.5). This strongly supports the practicality of our training-inference framework.
>
> In summary, while Dream and SDAR show that adaptation is possible, our work demonstrates that it can be done efficiently, robustly, and accurately with a carefully designed decoding strategy, training recipy, and system architecture.

---

> ### Author Response · Authors · 2025-11-22
> **Response to Reviewer 9tWZ (2/2)**
>
> ## Q3: The benefit of hierarchical cache
> ## A3:
> The hierarchical cache is a core part of our inference design, and its benefit comes from enabling two complementary forms of KV reuse within blockwise diffusion decoding:
>
> 1. Block-level caching (exact):
> Fully decoded blocks are cached and reused as fixed prefix context for subsequent blocks. This avoids any recomputation for past blocks and yields a stable speedup across all context lengths.
> 2. Sub-block caching (approximate):
> Within a block, we reuse KV states for already unmasked tokens during iterative refinement (via DualCache). This reduces intra-block overhead and accelerates the denoising steps.
>
> The hierarchical combination ensures efficiency at both levels:
>
> * In long-context scenarios, hierarchical cache maintains lower latency and higher throughput than block-only cache, as shown in the extended context experiments (1K–8K prefix lengths, figure 8 of the revised paper).
> * In a computation-bound setting, (e.g. large batch size), hierarchical cache further reduce the computation and improve the throughput without hurting accuracy (Figure 6).
>
> Importantly, hierarchical cache scales well in practice and, when combined with our parallel decoding strategy, achieves 2–3× overall speedup over strong AR baselines while preserving accuracy. This makes it a practical solution for real-world LLM deployment.
>
> ## Q4: The attribution of training cost reduction
> ## A4:
> We agree that both the block-wise architecture and the training recipe contribute to the observed 500× data efficiency improvement over Dream (1B vs. 580B tokens).
>
> Indeed, the block-wise dLLM structure is inherently more compatible with AR-style generation and likely reduces the data requirement compared to a full-attention diffusion model, such as Dream. However, our ablations (Table 2) show that using a blockwise architecture alone is insufficient — without our proposed improvements like complementary masking, and padding strategy, performance drops significantly.
>
> This suggests that the data efficiency in our approach stems from a tight integration of architectural design with a purpose-built adaptation strategy. The training recipe enables the model to make full use of limited data by aligning the learning dynamics with the blockwise decoding process.
>
> ## Q5: Conceptual comparasion with concurent work, SDAR
> ## A5:
>
> While SDAR is indeed a concurrent work that also explores finetuning AR models into block diffusion models, our approach differs in both training framework and inference pipeline:
> 1. Finetuning Framework:
> We introduce structured training techniques such as complementary masking, token shift prediction, and sub-block-aware attention schedules, which address challenges in block diffusion adaptation — especially under low-data regimes (~1B tokens).
> 2. Inference Pipeline:
> Our method supports sub-block decoding and a hierarchical caching architecture, enabling fine-grained generation and real-world latency improvements (Figures 5–6, Table 3).
>
> We also reference SDAR in our related work section to better position our contributions relative to this concurrent line of research.

---

> ### Author Response · Authors · 2025-11-27
> **A Gentle Reminder**
>
> Dear Reviewer 9tWZ,
>
> Thank you once again for your valuable comments! As the discussion phase is approaching its end, we would like to kindly confirm whether we have sufficiently addressed all of your concerns (or at least some of them). Should there be any remaining questions requiring further clarification, please do not hesitate to let us know. We sincerely look forward to your feedback.

---

### Official Review · Reviewer_3sof · 2025-11-01

**Soundness:** 3
**Presentation:** 3
**Contribution:** 2
**Rating:** 6
**Confidence:** 3

**Summary:**

This work introduces Fast-dLLM v2, which turns regular AR language models into faster diffusion models that can generate multiple tokens at once, using only ~1B tokens for fine-tuning. The key trick is a block-based approach where tokens within each block can look at each other while still respecting the left-to-right flow between blocks. With smart caching and parallel decoding, it runs up to 2.5× faster than standard models while keeping the same quality on benchmarks.

**Strengths:**

1. The method is elegantly simple yet effective. It adapts pretrained AR models into block diffusion models with minimal modifications, requiring only a complementary masking strategy and block-wise attention that can be easily integrated into existing training and inference frameworks.
2. The paper provides thorough experimental validation across multiple model scales and diverse benchmarks, including code generation, mathematical reasoning, and knowledge tasks, with comprehensive ablation studies examining the impact of block sizes, sub-block sizes, and caching strategies.
3. The presentation is easy to follow, with concrete experimental details that make the work solid.

**Weaknesses:**

1. Although the average performance remains the same compared with Nemo-FT models, there are consistent patterns in performance change in specific benchmarks. Eg. HumanEval has a significant increase, while Math experiences a large decrease in both 1.5B and 7B comparison, which is unlikely to be random fluctuations.
2. While causing some performance drops in some of the benchmarks, the proposed method only achieves 2.5x acceleration compared with AR models. The speed-up is similar to many existing acceleration methods for AR models, like speculative decoding, making the method less competitive.

**Questions:**

1. What could be causing the consistent and significant discrepancy mentioned in "Weakness"?
2. Can this block diffusion process be used together with existing methods to accelerate AR models? eg. speculative decoding.

---

> ### Author Response · Authors · 2025-11-22
> **Response to Reviewer 3sof**
>
> ## Q1: The consistent discrepancy in math and code benchmarks
> ## A1:
> Thank you for the observation — we agree that the performance changes across tasks are consistent and reflect meaningful interactions between task structure and decoding behavior.
>
> As shown in Table 1, Fast-dLLM v2 achieves significant gains on HumanEval, especially at the 7B scale, but shows reduced performance on Math tasks. This aligns with observations in prior diffusion-based LLMs, such as D2F[arxiv:2508.09192], where diffusion decoding showed strong acceleration benefits in code-related tasks, but relatively less gain in certain reasoning benchmarks (e.g., Math on LLaDA).
>
> We conjecture this is due to the structural prior:
> - Code tasks (e.g., HumanEval) benefit from structural block boundaries (e.g., indents, brackets), which fit well with blockwise decoding. Diffusion also favors string structure completion and can easily recover correct paths even when partially decoded.
> - Math tasks, on the other hand, involve strict, symbol-level logical chains, often requiring precise step-wise derivation. Global or bidirectional attention inside blocks may leak parts of the answer too early during training, hurting model learning or introducing shortcut bias.
>
> Importantly, the average performance still matches strong AR baselines, and we believe this is an opportunity for the future work to further go deep in it.
>
> ## Q2: Can this block diffusion process be used together with existing methods to accelerate AR models? eg. speculative decoding.
> ## A2:
>
> Thank you for the excellent question. We would like to clarify three key points:
> 1. Research Scope Difference:
> Our work focuses on improving the inference efficiency of diffusion-based LLMs, which is a parallel research direction to AR-based acceleration techniques such as speculative decoding. The 2.5× speedup we report is relative to strong AR baselines, and represents a direct gain from rethinking the decoding paradigm — not from tuning within AR methods.
> 2. Performance Context:
> While some benchmarks show moderate drops, Fast-dLLM v2 is finetuned with only 1.3B–3.1B tokens, which is significantly smaller than the training scale of top AR models. Despite this, Fast-dLLM v2 delivers competitive or stronger average performance across benchmarks (Table 1), especially at comparable scale.
> 3. Compatibility with AR Acceleration:
> Block diffusion is compatible with AR acceleration techniques like speculative decoding. Recent work such as DiffuSpec (arXiv:2510.02358) demonstrates that using a small diffusion model (Dream-7B) as the draft for speculative decoding can achieve 3.1× speedup with AR models like Qwen2.5-32B.
>
> Since Fast-dLLM v2-7B is nearly 10× faster than Dream-7B (see Figure 1), using it as a draft model in similar frameworks is likely to further enhance decoding speed, and is a promising direction for future work. We believe the combination of blockwise diffusion and AR acceleration methods provides a fruitful avenue for future research.

---

### Official Review · Reviewer_nsrh · 2025-11-01

**Soundness:** 3
**Presentation:** 2
**Contribution:** 3
**Rating:** 6
**Confidence:** 3

**Summary:**

The paper introduces a block-diffusion adaptation to convert a pretrained AR model into a block-wise parallel generation model, achieving up to 2.5× speed-up over standard AR decoding with minimal loss in accuracy.

Contributions include: introducing complementary masking during training, a "hierarchical" KV caching mechanism to enable efficient block decoding, and large-scale evaluations on models upto 7B in size.

**Strengths:**

- requires light weight finetuning to convert AR to diffusion models
- better and faster than DREAM style approaches, that utilizes full sequence diffusion for generation
- there is a configuration in Fast-dLLM v2 that offers speed ups compared to an AR model, with similar accuracy -- fulfiling the promise of diffusion models

**Weaknesses:**

My main concern is limited conceptual novelty: there is only a single change from the previous iteration of Fast-dLLM-v1, which is adding blockwise training and generation. This weakness is not a bad thing in itself.
Please look at the questions :)

Other weakness is: the cache is still an approximation.

**Questions:**

- i'm still confused on how the hierarchical caching mechanism works, could the authors provide more details?
- are different sub-blocks decoded autoregressively? or in a given block, different sub-blocks can be decoded in any order?
- why do you need sub-blocks? can't dual cache work within a block, without defining sub-blocks?
- how would the latency change when the context length increases? i think authors tested on sequence lengths upto 2K only, i'm very curious to see how the latency scales with increasing context length


Since i'm not an expert in this field, i'll be looking at feedback of other reviewers too to gain a better understanding.
That being said, i'm definitely willing to improve my score during our discussion :)

---

> ### Author Response · Authors · 2025-11-22
> **Response to Reviewer nsrh (1/2)**
>
> ## Q1: Conceptual novelty compared to Fast-dLLM v1
> ## A1:
>
> While the framework builds on Fast-dLLM v1, we would like to respectfully clarify that v2 introduces multiple orthogonal and impactful innovations beyond a single modification.
> 1.  We propose an AR-to-Diffusion adaptation framework (Section 3.2) that allows efficient reuse of pretrained AR models for blockwise diffusion decoding — achieving over 500× data efficiency (1B vs 580B tokens). This is facilitated by a new training strategy that combines complementary masking, token shift prediction, and block-diffusion styled attention mask, enabling diffusion-style decoding without retraining from scratch.
> 2. We design a hierarchical caching architecture (Section 3.3) that introduces two distinct caching levels: (i) block-level reuse across blocks, and (ii) sub-block-level KV reuse within blocks. This architecture supports flexible decoding granularity and significantly improves inference efficiency when inference meets the computation bound (see Figure 6).
> 3. We address the training-inference block size mismatch problem through sub-block decoding. This mechanism enables autoregressive, left-to-right intra-block refinement, maintaining consistency with training structure while offering fine-grained decoding. Ablations in Tables 3–4 verify its necessity.
>
> In addition to these architectural contributions, Fast-dLLM v2 significantly improves performance over Fast-dLLM v1 and related baselines. As shown in Table 1, Fast-dLLM v2 outperforms Dream (trained on 580B tokens, served as the backbone of Fast-dLLM v1) by a notable margin in average score (60.3 vs. 57.6), while requiring dramatically fewer training resources (1B tokens only).
>
> To summarize: Fast-dLLM v1 focused on caching for full-attention diffusion models, while Fast-dLLM v2 introduces a complete blockwise diffusion system with (i) pretrained AR adaptation, (ii) hierarchical caching, and (iii) sub-block decoding — each contributing distinct advances. We will emphasize these points more clearly in the revised manuscript.
>
> ## Q2: Cache is still an approximation.
> ## A2:
> Our cache consists of two levels with different properties:
> 1. Block-level cache (inter-block, exact):
> Since blockwise diffusion generates blocks autoregressively and each block only attends to previous ones, once a block is decoded, its KV can be cached exactly. This cache is reused as a fixed prefix and introduces no approximations.
> 2. Sub-block cache (intra-block, approximate):
> Within a block, we use DualCache (from Fast-dLLM v1) to avoid recomputation during iterative decoding. This intra-block cache is approximate, as attention may change slightly between steps.
>
> In practical scenarios, a block-level cache is adequate as long as computation is not the limiting factor (see Figure 6). When paired with a sub-block cache, it can boost throughput in situations where computation becomes the bottleneck.
>
>
> ## Q3: Hierarchical caching mechanism
> ## A3:
> We are happy to clarify the hierarchical caching mechanism in Fast-dLLM v2, which consists of two levels:
> * Block-level cache (inter-block):
> After decoding a block (e.g., Block i), its key/value (KV) states are cached and reused as a fixed prefix for the next block (Block i+1), with causal attention. Since each block only attends to previous blocks, this cache is exact and requires no recomputation. This significantly reduces computation when generating long sequences.
> * Sub-block cache (intra-block):
> Within each block, decoding proceeds iteratively using diffusion steps. We apply DualCache (from Fast-dLLM v1) to avoid recomputing all token interactions at every step. It maintains:
>   * A prefix cache for decoded (unmasked) tokens
>   * A suffix cache for remaining masked tokens
>
> This is a local approximation, but accelerates decoding within a block without hurting accuracy (as shown in Figure 6).
> Together, these two levels form a hierarchical cache — exact across blocks and approximate within. This design allows Fast-dLLM v2 to scale efficiently while maintaining generation quality.

---

> ### Author Response · Authors · 2025-11-22
> **Response to Reviewer nsrh (2/2)**
>
> ## Q4: The schedule of sub-block decoding
> ## A4:
> In our approach, sub-blocks are decoded following a left-to-right schedule, but they are fully visible to each other through bidirectional attention during decoding.
>
> More specifically:
> - All sub-blocks within a block (e.g., 4 sub-blocks in a 32-token block) attend to each other symmetrically as well as all the previous blocks during diffusion decoding, just like the full block would.
> - We decode them sequentially — starting with sub-block 0, then sub-block 1, and so on — based on token confidence.
>
> To make this process more intuitive, we have included a visualization of the decoding process in our supplementary material — in the form of an animated GIF that illustrates how sub-blocks are progressively decoded within a block. The animation shows both the denoising dynamics and the left-to-right sub-block schedule over time. We hope this helps clarify how attention visibility and decoding schedule interact in practice.
>
> ## Q5: The need for sub-blocks
> ## A5:
> Sub-blocks are necessary from both a performance and efficiency perspective.
> - Performance (Table 3): Without modifying the model architecture, using sub-block decoding as a left-to-right schedule leads to better generation accuracy. This is because sub-blocks allow the model to condition on growing portions of the decoded prefix within a block. As shown in Table 3, using sub-blocks of size 8 achieves significantly better performance on GSM8K and HumanEval compared to decoding the full block in a single pass.
> - Efficiency: From a systems perspective, sub-blocks introduce explicit structure that enables granular application of DualCache. Instead of recomputing the entire block, the model can cache intermediate prefix/suffix KV states at the sub-block level — allowing fast partial recomputation with minimal overhead. This hierarchical structure is far more practical than trying to apply DualCache over a monolithic, unstructured block.
>
> In short, using DualCache without sub-blocks would either (a) lose performance due to unorgranized decoding schedule, or (b) require inefficient handling of cache without proper structure. Sub-blocks strike the right trade-off between accuracy and efficiency, and are essential to the overall design of Fast-dLLM v2.
>
> ## Q6: The latency changes when the context length increases
> ## A6:
> To explore this, we conducted an experiment varying the context length from 1K to 8K tokens, and measured both generation latency and throughput for generating 32 completion tokens. The tests were run on a single NVIDIA A100 80GB GPU.
> As shown in the figure 8 of the revised paper (Generation Latency / Speed Comparison), our hierarchical cache consistently outperforms block-only cache, particularly at longer contexts. Notably, it also achieves latency and throughput comparable to standard AR model caching, showing that our caching system is highly effective in reducing overhead under blockwise diffusion decoding.
>
> Furthermore, when combined with our parallel decoding strategy (i.e., sub-block speculative denoising), the overall method achieves 2–3× speedup over strong AR baselines — as shown in Figure 1 of the main paper.

---

### Author Response · Authors · 2025-11-27
**General Response to All Reviewers**

We sincerely thank all reviewers for their constructive feedback. In our individual responses, we have addressed the critical concerns regarding novelty, mechanics, and validation.

Summary of Updates:

* Novelty & Contribution (Response to R-nsrh, R-9tWZ): We clarified that Fast-dLLM v2 introduces orthogonal innovations (hierarchical caching & sub-block decoding) beyond v1/concurrent works, solving key bottlenecks in inference latency.
* Performance & Generalizability (Response to R-3sof, R-aB3q): We expanded our evaluation to include experiments on Llama architectures, confirming our method's robustness across different base models. We also addressed domain-specific trade-offs (e.g., Math/Code).
* Mechanism Details (Response to R-nsrh): We provided comprehensive breakdowns of the scheduling strategy and cache operations.

We hope these additional experiments and analyses have resolved the concerns raised. We would appreciate it if you could let us know if there are any remaining questions. If our response has adequately addressed your comments, we kindly hope you might consider raising your score.

Best regards,

The Authors

---

### Meta-Review · Area_Chair_Rzf6 · 2026-01-07

**Summary:**

Reviewers agreed that the paper tackles an important problem—LLM inference latency—and presents a practically strong system that adapts pretrained autoregressive models into efficient block-diffusion models using minimal fine-tuning data. The main concerns centered on the degree of conceptual novelty relative to prior and concurrent work, clarity and necessity of the hierarchical caching and sub-block decoding mechanisms, performance trade-offs across different task types, and how much of the reported data efficiency stems from architectural choices versus the proposed training recipe. The authors’ rebuttal provided detailed clarifications, new experiments, and expanded analyses that addressed many of these concerns, strengthening confidence in the technical soundness and practical contribution of the work.

**Reviewer Concerns:**

Reviewers raised concerns about whether the contributions represent incremental engineering rather than conceptual novelty, noting overlap with prior work such as Fast-dLLM v1. Additional questions focused on the necessity and design of the hierarchical caching mechanism, the role of sub-block decoding, scalability to longer context lengths, and observed performance discrepancies across benchmarks (e.g., gains on code tasks but drops on some math tasks). In the rebuttal, the authors carefully decomposed the contributions, clarifying how Fast-dLLM v2 differs from prior work. They added new experiments on longer contexts, additional architectures, and detailed ablations showing when and why sub-blocks and hierarchical caching are necessary. While some reviewers may still view the work as primarily systems-oriented rather than conceptually novel, the rebuttal resolves most technical and empirical concerns, and no critical issues remain.

**Reviewer Scores:**

Given the additional explanations and experiments, it is likely that borderline scores would move upward or be held with greater confidence, while strong positive scores would remain unchanged.

---

### Decision · Program_Chairs · 2026-01-26

Accept (Poster)